# Family planning demand generation in Rwanda: Government efforts at the national and community level impact interpersonal communication and family norms

Julia Corey[1]*, Hilary Schwandt[2], Angel Boulware[3], Ana Herrera[4], Ethan Hudler[5], Claudette Imbabazi[6], Ilia King[7], Jessica Linus[8], Innocent Manzi[6], Madelyn Merrit[9], Lyn Mezier[10], Abigail Miller[2], Haley Morris[11], Dieudonne Musemakweli[6], Uwase Musekura[12], Divine Mutuyimana[6], Chimene Ntakarutimana[13], Nirali Patel[14], Adriana Scanteianu[15], Biganette-Evidente Shemeza[6], Gi'anna Sterling-Donaldson[16], Chantal Umutoni[6], Lyse Uwera[6], Madeleine Zeiler[2], Seth Feinberg[9]

1 Wheaton College, Norton, Massachusetts, United States of America, 2 Fairhaven College, Western Washington University, Bellingham, Washington, United States of America, 3 Spelman College, Atlanta, Georgia, United States of America, 4 Northwest Vista Community College, San Antonio, Texas, United States of America, 5 Whatcom Community College, Bellingham, Washington, United States of America, 6 INES, Ruhengeri, Musanze, Rwanda, 7 Xavier University, New Orleans, Louisiana, United States of America, 8 UMBC, Baltimore, Maryland, United States of America, 9 Department of Sociology, Western Washington University, Bellingham, Washington, United States of America, 10 SUNY Oswego, Oswego, New York, United States of America, 11 Western Oregon University, Monmouth, Oregon, United States of America, 12 Eastern Oregon University, La Grande, Oregon, United States of America, 13 University of Kentucky, Lexington, Kentucky, United States of America, 14 Arcadia University, Glenside, Pennsylvania, United States of America, 15 Rutgers, New Brunswick, New Jersey, United States of America, 16 Drexel University, Philadelphia, Pennsylvania, United States of America

* corey_julia@wheatoncollege.edu

## Abstract

Between 2005 and 2020, total contraceptive use among married women in Rwanda increased from 17% to 64%. The aim of this study is to better understand how the Rwandan government's mobilization and demand generation efforts have impacted community norms and interpersonal discourse surrounding family planning. Eight focus group discussions with family planning providers and 32 in-depth interviews with experienced modern contraceptive users were conducted in 2018 in the two Rwandan districts with the highest and the lowest contraceptive prevalence rates. Results suggest that outspoken government support, mass media, and community meetings were valuable sources of information about family planning. Information received through these channels generated interpersonal dialogue about contraceptives through both conversation and observation; however, rumors and misinformation remained a significant barrier to use. A once taboo subject is now normative among married couples. Continuing to address common fears and misinformation through communication channels such as mass media and community meetings may help to further increase contraceptive uptake in Rwanda.

**Data Availability Statement:** There are ethical restriction on sharing the data publicly, as we did not include this issue in our human subjects review

documents nor in our consent forms. In addition, the data include personal and sensitive topics on individual's sexual and reproductive health experiences. However, the data are available on reasonable request from Janai Symons, the Research Compliance Officer at Western Washington University at 360-650-3082 or lanej4@wwu.edu.

**Funding:** The funding for this research came from the National Science Foundation (NSF) under the Research for Undergraduates (REU) funding stream, grant number 1852411. The funders had no role in study design, data collection and analysis, decision to publish, or preparation of the manuscript. Two of the authors received two weeks of their salary from this NSF grant, they are: Hilary Schwandt and Seth Feinberg.

**Competing interests:** The authors have declared that no competing interests exist.

## Introduction

218 million women of reproductive age (15–49) in low and middle-income countries (LMICs) who do not wish to become pregnant are not using a modern contraceptive method [1]. Estimates from 2019 suggest that 111 million unintended pregnancies, 30 million unplanned births and 35 million unsafe abortions take place in LMICs every year [2]. If unmet need for modern contraception was satisfied for all women in LMICs, and all pregnant women received adequate care, it is approximated that unintended pregnancies would be reduced by 68%, unplanned births by 71%, and unsafe abortions by 72%. Additionally, maternal mortality would be reduced by 62%, preventing an estimated 113,000 maternal deaths annually [2].

One approach to increasing uptake of modern contraceptives is generating demand through communication programs. Communication drives change in knowledge, attitudes, and social norms at the individual, community and national policy levels [3, 4]. According to Roger's [4] diffusion of innovations theory, "diffusion is the process by which an innovation is communicated through certain channels over time among the members of a social system." Once individuals begin using an innovation like family planning, communication about experiences spread throughout social networks. Adoption typically starts slow, but as more individuals begin using contraceptive methods, presence of information and peer-influence increases and demand for and use of family planning takes off, until eventually reaching saturation and levelling off. As such, family planning communication programs use mass media and community mobilization as channels to promote interpersonal dialogue and increase demand and uptake of contraceptives [5, 6]. Using mass media to promote family planning enables programs to reach large populations and share ideas in an entertaining manner within a relatively short period of time [7–10]. This helps foster dialogue, increase social approval, and improve overall knowledge and perceptions about family planning [10]. Exposure to messages about family planning in the media is positively associated with contraceptive use [7, 9–12].

In 2000, the Rwandan government introduced 'Vision 2020', a development programme that aimed to make Rwanda a middle-income country [13]. As one of the most densely populated countries in Africa [14], reducing the fertility rate was seen as key to reducing infant and maternal mortality, and producing a vibrant and educated middle class that could develop the country [13]. As such, the government launched a massive, multisectoral public education campaign promoting family planning as a tool to reduce poverty and develop the country [13, 15], aiming to reduce the fertility rate from 6.5 to 4.5 total births per woman by 2020 [13]. Messages encouraging families to use contraceptive methods to have fewer children and increase birth spacing are broadcast throughout Rwanda using media such as the radio, television, newspapers, and magazines [16]. In 2020, 49% of women and 63% of men of reproductive age reported seeing or hearing family planning messages on the radio in the few months prior to being surveyed [17], an increase from 35% and 50%, respectively, in 2000 [18], while household radio ownership increased only 5% over the same period. At the community level, local authorities are required to mention reproductive health every time they address their constituencies. Additionally, during *Umuganda*, Rwanda's monthly community service day, both parliamentarians and community health workers (CHWs) share important information with communities, including family planning [19].

Between 2005 and 2010, the percentage of married women in Rwanda using any contraceptive method more than tripled from 17% to 52% and then increased to 64% in 2020 [20–22]. Use of any contraceptive method also increased by 50% among rural women and 47% among those with no education between 2005 and 2020 [21–23]. Unmet need for family planning among married women decreased from 38% [13] to 14% [22] during the same period. As a result, Rwanda's fertility rate dropped below 4.5 total births per woman in 2011 [24], nine

years ahead of the government's goal, and has continued to decline, reaching 4.1 total births per woman as of 2020 [22].

It is important to understand how family planning information is communicated to women and couples of childbearing age, and how this affects community norms and interpersonal discourse. Studying these sociocultural influences may offer insights as to why total contraceptive use among married women in Rwanda increased rapidly. Doing so may enable scholars and practitioners to consider ways that may help continue to increase contraceptive prevalence in both Rwanda as well as other regions of the world most likely to feel the acute impacts of unmet family planning needs. This study aimed to better understand how the government's mobilization and demand generation efforts have impacted community norms and interpersonal discourse surrounding family planning in Rwanda.

## Methods

Data were collected through focus group discussions (FGDs) and semi-structured in-depth interviews (IDIs) in the Musanze and Nyamasheke districts of Rwanda. Musanze has a total fertility rate of 3.5 births per woman compared to 4.7 in Nyamasheke, and Musanze is 28% urban as compared to 2% in Nyamasheke [22]. These districts also represent the areas with the highest and lowest rates of modern contraceptive prevalence, respectively, which made them ideal for this study given the focus on contraceptive use and an interest in contraceptive use, as well as any difference in reasons for use, or non-use, by district.

A total of 88 family planning providers participated in eight FGDs–four FGDs with family planning nurses and four FGDs with CHWs–split equally between the two districts in February 2018, in order to have at least 2 FGDs by sample characteristic, in this study: district and provider type. Family planning nurses and CHWs were recruited via NGO and government staff working in positions knowledgeable about and familiar with all family planning providers in the districts. Additionally, 32 IDIs (16 in each district) with experienced modern contraceptive users were conducted in July 2018. NGO and government staff, and providers who participated in the study, assisted in recruitment of contraceptive users.

All FGDs and IDIs were conducted in Kinyarwanda and audio recorded by male and female native Kinyarwanda speakers with undergraduate education. The data collectors were recruited by our study collaborator, their university professor. Training included an overview of the study, review of the topic guides, best practices for qualitative data collection, role of the data collectors, data analysis, and research ethics. FGDs were conducted once per day, in private rooms at public institutions, and 4 IDIs were conducted each day by 4 different interviewers in outdoor, private spaces agreed upon by the participant and interviewer. Only study participants and data collectors were present for data collection, and personal identifiers were not included in data collection. All participants provided written informed consent. Ethical approval was obtained by the Institutional Review Boards at Western Washington University and the Ministry of Education in Rwanda.

All audio recordings and field notes were translated and transcribed into English by the interviewers, working alongside native English speakers, immediately following data collection. Audio recordings and transcriptions were stored on password protected computers. On average, FGDs lasted approximately two hours, and IDIs about 43 minutes. Data coding was guided by thematic content analysis [25] until inductive thematic saturation was reached [26] using Atlas.ti 8 [27] and group level matrices in Microsoft Excel.

## Results

Family planning providers and users highlighted the theme of "mobilization," which they described as the encouragement to use contraceptives at national, community, and

interpersonal levels. These mobilization efforts were discussed in a positive manner and were described as a key contributing factor to the dramatic increase in contraceptive use throughout the country. Both providers and users discussed strategies such as national media campaigns and community educational events as examples of mobilization efforts designed to increase demand for contraceptive use and combat misinformation. Participants added that mobilization at the national and community levels can generate and influence interpersonal dialogue.

## Motivation for mobilization

Study participants described two motivations for mobilization: (1) to increase awareness and demand for contraceptives and (2) to dispel negative rumors about contraceptives. The topic of rumors arose in every focus group discussion with family planning providers and were related to the side effects a woman may experience with hormonal contraceptive use. Providers noted that fears can sometimes lead to discontinuation of contraceptive use, in some cases even when the user has not personally experienced any side effects.

> She might want to change to a different method depending on bad rumors she got from another woman. Other women may tell her that if you take pills you will have a bad menstruation, and that it will cause other bad effects on your body, so she will get afraid. This will cause her to stop using the pills before any bad effects even show up.
>
> –CHW, male, 46, Musanze, FGD

The theme of rumors also arose as part of the IDIs with family planning users. Participants similarly brought up that fears about potential side effects can discourage both new and continuing users from using family planning. Most commonly, women reported rumors that contraceptives caused sterilization. Contributions to this theme were made three times more often by participants in Nyamasheke than in Musanze.

> The first rumors I heard were when I was pregnant with my first child. They caused me to not go in after I gave birth and get contraceptives.
>
> –Female, 35, condom user, 4 children, Nyamasheke, IDI

Providers were clearly challenged by rumors and were working hard to combat them with their knowledge and experience, but felt they were not always able to overcome this challenge.

> Most women understand rumors more than truth.
>
> –Nurse, female, 42, Musanze, FGD

## Mobilization at the national level

**Government leadership.**   Family planning providers in every FGD spoke positively about the Rwandan government's involvement in the family planning program through both national policy and intervention. Both CHWs and nurses emphasized that the government's endorsement of family planning has been key to increasing contraceptive use. Most family planning users also perceived the government to be working effectively to develop and improve Rwanda's family planning program.

> The easiest aspect of our job is the citizens already know that the family planning program is supported by the government. So, it is easy for us to go into the villages and provide these services without any doubts.
>
> –CHW, male, 40, Nyamasheke, FGD

**Media campaigns.** The Rwandan government runs media campaigns through various outlets to promote and sensitize people to the idea of contraceptive use. Messages include information about how family planning can help develop and better both families and the country as a whole. Providers and users believed that education and consistent messaging on this national scale helps to increase contraceptive use through increased awareness and dialogue at the interpersonal and community levels.

> . . .they use the radio and the television and placards to talk about family planning. They put everywhere the information about family planning.
>
> –Nurse, female, 39, Nyamasheke, FGD

Contraceptive users discussed government messaging about family planning, and two thirds specifically mentioned hearing messages on the radio. Several participants reported that the information they heard was in relation to positive family management and developing the country. Exposure to family planning messages on the radio, along with exposures in other forms, were often the first exposure to family planning and a catalyst for women to seek out providers to discuss initiating contraceptive use. While the majority of participants mentioned hearing the family planning messages on the radio, it was noted more often in Musanze than in Nyamasheke.

> I heard the information on the radio and that's when I got the curiosity. And when the time came that I thought it was a good time to start using contraceptives, I went to the CHW and they explained to me how they work. Then I started using contraceptives.
>
> –Female, 41, injectable user, 5 children, Musanze, IDI

As this participant explained, exposure to information about family planning is well done in Rwanda.

> Interviewer: How did you first get information about family planning?
>
> Respondent: I can't say the exact place because this program is already known.
>
> –Female, 29, implant user, 2 children, Nyamasheke, IDI

## Mobilization at the community level

At the community level, Rwanda's family planning program provides formal spaces for education, sensitization, and testimonials to be shared. Family planning providers shared that community education has been integral to the success of the program and described it as an impactful aspect of their jobs. The sense was that community education is a tool that is used to normalize and encourage use of family planning, dispel rumors, provide a platform for testimonies, and mobilize people to seek contraceptives if they desire. Providers felt these events were important because they can serve as an entry point for women who previously have not considered using family planning.

Contraceptive users also discussed these community meetings as important sources of information for both the community and individuals. Women described community meetings as places where people learn about the importance of family planning, how to use it, various benefits, as well as how to manage any issues related contraceptive use. Over 70% of respondents reported receiving family planning information from community meetings, most commonly *Akagoroba K'ababyeyi* (59% of respondents) and *Umuganda* (34% of respondents).

*Akagoroba K'ababyeyi*, or "women's evening" is a space for mothers to come together and share informal stories and experiences about important issues such as malnutrition, education, family planning, and other important topics for parents. These meetings take place monthly within villages to ensure the space is easily accessible to community members. Providers mentioned that these events promote community and interpersonal support and are especially helpful to women who do not have family members or neighbors who they can talk openly with about family planning.

> There are programs where every woman in the village meets at night to discuss about everything (Akagoroba K'ababyeyi). During the meeting they get information about how to use family planning. . .women who use contraceptives give them a testimony. They knew her before she used a contraceptive, and after, so they can see the benefits she has had from using contraceptives. The testimonies help them.
>
> –Nurse, female, 34, Musanze, FGD

*Umuganda*, or "coming together in common purpose", is Rwanda's monthly day of mandatory national service and was the community event that was discussed most frequently among CHWs. Nurses also brought up *Umuganda*, but less often. Community meetings that are held following *Umuganda* are a space for formal education about family planning—when CHWs and nurses discuss what services are available and where they can be accessed. The majority of providers who brought up *Umuganda* discussed how they are used to help sensitize the community to becoming more comfortable discussing and seeking family planning services. These meetings were also important to respondents because they helped dispel rumors about family planning that existed within the community.

> It is very important that we teach about family planning during Umuganda because it helps people become open to asking for help.
>
> –CHW, female, 47, Musanze, FGD

### Interpersonal communication

The combination of national efforts through mass media and the support from the communities through events helps to shift the social norms in Rwanda and foster communication about family planning on an interpersonal level. Both providers and users acknowledged the impact that social networks and peer-to-peer communication has had on generating demand for contraceptives.

**"Do it well and others will see".** The use of examples to inspire uptake of contraceptives was a common theme among family planning users, and various examples were discussed: from themselves to family planning providers, and their neighbors. Women noted how they look to their community members for both good and bad examples of the outcomes of contraceptive use or non-use. Through observation, women can notice birth spacing between another woman's children, motivating them to think about initiating conversations, seeking assistance, and potentially beginning to use family planning.

First, I noticed it through my neighbor who had a child of ten years and I asked her how it was possible to have a child of ten years and she has not been pregnant since. That is when I approached her and asked how she did that and she told me about family planning. . .

–Female, 26, injectable user, 1 child, Musanze, IDI

Contraceptive users also noted that neighbors could serve as negative examples by demonstrating outcomes of not using contraceptives. Seeing neighbors struggle with unplanned children motivated some women to both initiate and sustain family planning use.

You can find a woman who has an eight-month-old child at home and is five months pregnant at the time and that serves as an example for us. When you see how this woman is suffering, having to live in this way, you continue using family planning.

–Female, 32, injectable user, 3 children, Musanze, IDI

Women sometimes compared the results of their decision to use family planning to others in their communities who did not decide to use contraceptives.

If I did not take the decision to use family planning, by now I would have had three children. I know someone who gave birth during the same year as me and that woman now has two other children and is struggling to raise her children. If that was me, those children would have had a problem of malnutrition and I would have not been able to take care of them. Also, I gained the ability to save money because I do not have a big family.

–Female, 26, injectable user, 1 child, Musanze, IDI

Participants mentioned wanting to set good examples of family planning use for others in their communities to motivate them to use contraceptives as well. Women in the IDIs felt that providers must also serve as positive examples of family planning use. This is likely even more important in Rwanda where CHWs are elected by and live in the same communities they serve.

. . .if some woman sees that a family planning services provider is not using family planning, she can say that 'oh you are giving advice but you are not using family planning.' Because there is a proverb in Rwanda saying that 'do it well and others will see' (*nkore neza bandebereho*).

–Female, 38, implant user, 5 children, Nyamasheke, IDI

**Peer-to-peer motivation and support.** Family planning users were asked if they discuss family planning with their friends, co-workers, or neighbors: 90% of respondents said they discuss with their friends, 25% with neighbors, and 21% with co-workers. Contraceptive users often explain to their friends the benefits they have experienced themselves through using methods. Benefits that women mentioned included feeling stronger, being able to provide for their family, and being able to work.

I usually discuss with my friends how good family planning is because you don't get easily tired. You have children when you are ready and able to take care of more. . . We also discuss the benefits of family planning and also the experiences we've all had with it. We advise each other based on the method we are using.

–Female, 38, condom user, 4 children, Nyamasheke, IDI

Some contraceptive users reported that their first source of information about family planning was their friends, or that their friends were the ones who motivated them to seek out contraceptives. Additionally, some women even chose their method based on what their peers have used. Personal testimonies about specific contraceptive methods were a powerful guide for other women in making their own method choices.

. . .my friends didn't have any bad experiences when using injectables, so I thought that it would make sense for me to use injectables. . .

–Female, 38, condom user, 4 children, Nyamasheke, IDI

Providers recognized that the relationships they have with clients can extend beyond that individual to other potential contraceptive users. Providers noticed that if a woman listens to her peers and decides to use family planning, she will eventually share her own positive experience with others, leading to a demand generation effect. Contraceptive users also noted that the advice they share with one woman can continue to spread further.

If you receive her kindly, she will go to tell her friends and this can encourage them to come.

–Nurse, female, 38, Nyamasheke, FGD

Contraceptive users also explain the different methods to their friends, and what they can expect with their bodies when using family planning. Sometimes this helps their friends worry less because they will know what to expect.

. . .I would tell her that using contraceptives is important. It may have side effects or some problems but you can go see a doctor and the doctor will give you advice about how you can handle those problems.

–Female, 37, condom user, 2 children, Nyamasheke, IDI

Providers noted that this interpersonal communication and mobilization can sometimes help combat rumors or false information that can spread about contraceptives because of stigma.

## Shifting family norms

Whereas in the past ". . .it was a secret, nobody was sharing family planning because many people were judged. . ." (Female, 26, injectable user, 1 child, Musanze, IDI), many family planning users in this study noted that contraceptive use is normalized among women in their communities today. This was mentioned more often in Musanze than in Nyamasheke.

The friends I have right now are on the same page as me; they all use contraceptives.

–Female, 38, pill user, 3 children, Musanze, IDI

Participants discussed how the changing perceptions of family planning use were initially received poorly, but later embraced as a result of increased awareness, support, and acceptance of contraceptive use.

. . .In the past they used to teach us, and we didn't see the importance of using contraceptives. But now because there is a better awareness, no one has a big issue with using contraceptives. All people understand why it is important.

–Female, 41, injectable user, 5 children, Musanze, IDI

## Discussion

This study sought to better understand the Rwandan government's contraceptive demand generation efforts and how they have impacted community norms and interpersonal discourse surrounding family planning. The results demonstrate that interventions at both the national and community level serve as catalysts for interpersonal communication. This was revealed through discussions about government leadership, mass media campaigns, accessibility, community meetings, peer-to-peer support, and social norms.

In Rwanda, media campaigns serve as a channel of communication to share knowledge and information about family planning. The radio was by far the most commonly reported source of family planning information in the media, as has been noted in population-based surveys [17]. Given that Nyamasheke has some of the highest levels of poverty in the country, it was not unexpected that hearing family planning messages on the radio was reported less often by women in this district than in Musanze. In Rwanda, an estimated 40% of households have a radio, and nearly half of women ages 15–49 reported hearing family planning messages on the radio in 2020 [21]. In neighboring countries Tanzania and Uganda, where 49% [28] and 59% [29] of households possess a radio, respectively, similar findings have been reported; 62% of women in Tanzania [30] and 65% of women in Uganda [31] reported hearing family planning messages on the radio in 2016. Hearing messages on the radio was often one of the first exposures to family planning for women in this study. Information heard through the mass media was a catalyst to seek further information about family planning from providers. This, in line with a study in Nigeria by Bankole [7], supports the diffusion of innovations theory in that mass media is important to the knowledge stage of the innovation decision process and can generate dialogue about family planning.

While Rogers [4] focuses on media and interpersonal dialogue as the main channels of communication, this study found that community meetings in Rwanda serve as another important platform for providing information and promoting communication about family planning. Formal community meetings such as those held following *Umuganda* are a space for providers to address the entire community with information about family planning, including the available services and where they can be accessed. They also help to dispel rumors circulating in the community and normalize dialogue about contraceptives. Discussing family planning in the community setting has also helped to promote male involvement and positive spousal communication [32]. Community education has played an important role in the demand and uptake of contraceptives in Rwanda as well as in Nigeria [10].

While providers and formal announcements serve as an important source of knowledge and information, most individuals look to their peers for subjective experiences and opinions surrounding family planning use in order to form their own opinions and decisions [4]. This may be why *Akagoroba K'ababyeyi* was mentioned more often than any other community meeting by modern contraceptive users in this study; the informal sharing of personal testimonies from women like themselves enabled them to see how contraceptives work and the various benefits. This finding is supported by a study in Bangladesh that saw an increase in modern contraceptive uptake as a result of peer support meetings to share testimonials about family planning [33].

Nearly all of the modern contraceptive users in this study mentioned discussing family planning with their peers, demonstrating that interpersonal dialogue is an important channel of communication. Peers are an important source of information; they share both their positive and negative experiences with contraceptive methods and providers. These personal testimonies are powerful and can influence a woman's decision to use or not use contraceptives, as well as which method to choose. Similar findings were reported in Malawi [34].

Observational learning has a powerful effect on health behavior [35, 36]. In Rwanda, examples are another catalyst for interpersonal communication and demand generation. Seeing other women achieving birth spacing sparks curiosity and initiates conversations about family planning, which can lead to contraceptive uptake. Likewise, seeing women have many children with no birth spacing serves as an example of the negative outcomes of not using family planning, and motivates contraceptive users to sustain use. Similarly, a study in Malawi reported that observing others struggle with children close in age sparked conversations about contraceptives among women, and men knew whether their peers used family planning based on observed birth spacing [34]. Perceived use of contraceptives among peers can have a powerful effect on uptake; Calhoun et al [37] reported that sexually active young men and women in Kenya who perceived their peers to be using modern contraceptive methods were significantly more likely to report using a method themselves.

While contraceptive methods are well-known and widely available in Rwanda [17], rumors and misinformation remain a significant barrier. Specifically, fears of side effects and health concerns are common reasons for both non-use and discontinuation of family planning, sometimes even among those who have not experienced them personally. Worldwide, this holds true [38, 39]. While some fears of side effects are based on truth, many arise from rumors [40, 41]. Studies have suggested that when these rumors are addressed and dispelled, women are more open to using family planning [42]. Being that family planning is well-known in Rwanda today, focusing media and communication efforts toward addressing common fears and misinformation may help further increase uptake to meet total demand and reduce unmet need. This may be especially important in Nyamasheke, given that the subject of rumors was brought up three times more often by participants in this district than in Musanze and that it is the district with the lowest contraceptive prevalence rate.

This study found that the government endorsement of family planning has been key to the success of the program. These findings are supported by Zulu et al. [19], who add that the Rwandan people have high respect for authority and leadership, and therefore follow decisions made at the community, regional, and national levels. Family planning providers have also been impactful. Both nurses and CHWs are well-respected members of their communities and play an important role in the decision to use contraceptives. Providers' attendance and advocacy for family planning at community meetings helps to sensitize communities. At the interpersonal level, providers share accurate information, clarify misguided rumors, and help clients find a method that will work well for them. This is especially important to sustaining contraceptive use among dissatisfied users, as continued counselling helps to prevent discontinuation [43].

In years past, family planning was a taboo topic in Rwanda and women were expected to have many children. Following the 1994 genocide, desire to rebuild the population and replace those who were lost made promotion of family planning to those who lost loved ones even more difficult [19]. The promotion of family planning as a tool for improving families and developing the entire nation, rather than as just a medical issue, has resonated across the country [19]. Today, contraceptive use among married couples is normalized. Focus has shifted from having as many children as possible to instead having the number of children that one can provide for [44]. While the topic of normalization arose among participants in both

districts, it was more frequently noted by women in Musanze. This was unsurprising, given that Musanze is the district with the highest contraceptive prevalence rate.

While the family planning program in Rwanda has been highly successful, there are areas in which it may be improved. The government should continue to promote the program via mass media; messages should be tailored to include more information addressing common misinformation and fears among the population. Additionally, community visits from government representatives explaining the benefits of family planning may be particularly useful in areas where modern contraceptive prevalence remains low. Further investment in human resources such as nurses and CHWs would also be valuable, to improve accessibility and ensure that every household is reached [43].

This study has several strengths. Data were collected from two districts in Rwanda–the district with the highest contraceptive prevalence (Musanze) and the district with the lowest contraceptive prevalence (Nyamasheke). Doing so highlighted some of the key differences in how communication about family planning occurs in each district and may provide insights as to why the contraceptive prevalence rates differ. Additionally, study participants included experienced modern contraceptive users as well as family planning providers, drawing from both nurses and CHWs. Inclusion of these various stakeholders, all with diverse perspectives, provided a multifaceted understanding of Rwanda's family planning program. Despite a number of strengths, this study has a few limitations. Audio recordings were translated and transcribed directly into English. Had recordings been first transcribed in Kinyarwanda, and then translated into English, translation may have been more accurate. Additionally, experienced modern contraceptive users were recruited by family planning providers, so it is likely that selection bias occurred, with recruitment of participants who have had more positive experiences with family planning than negative.

This study included interviews with experienced modern contraceptive users. To better understand barriers to demand and uptake of family planning in Rwanda, future studies should seek women who have never used and have no intentions to use contraceptives to better understand barriers and facilitators to uptake. It would also be valuable to examine communication channels among adolescents and unmarried youth and adults, who were not included in this study. Doing so may inform ways in which the family planning program in Rwanda can continue to improve and overcome barriers to contraceptive uptake for these groups.

## Conclusion

This study aimed to better understand how the Rwandan government's mobilization and demand generation efforts have impacted community norms and interpersonal discourse surrounding family planning. Rumors and misinformation about contraceptives that are shared within social networks can sometimes lead to fears that prevent women from using or continuing family planning. This was noted as a significant barrier to contraceptive use. Efforts at the national level, such as outspoken government support and media campaigns, generate dialogue about contraceptives among individuals. Community meetings also serve as an important platform for sharing family planning information and testimonies. At the interpersonal level, communication about family planning occurs through both conversations and observational learning. Peers serve as an important source of information, share personal experiences with one another, and provide support. Through observation of others, women learn both the benefits of using contraceptives, and the consequences of not using family planning. Outspoken government support and promotion of family planning through communication channels has generated positive dialogue on a topic that was previously considered taboo. Whereas

women were previously expected to have many children, focus has now shifted to having the number of children one can provide for in order to develop both the family and the country as a whole. Continuing to address common fears and misinformation through communication channels such as mass media and community meetings will likely help to further increase total contraceptive uptake and reduce unmet need in Rwanda.

## Acknowledgments

The authors wish to acknowledge Dean Faustin Habineza and the academic leadership at INES for their role in facilitating this international research collaboration. We would also like to acknowledge all of the persons who participated in and contributed to our research study.

## Author Contributions

**Conceptualization:** Hilary Schwandt.

**Data curation:** Claudette Imbabazi, Innocent Manzi, Dieudonne Musemakweli, Divine Mutuyimana, Biganette-Evidente Shemeza, Chantal Umutoni, Lyse Uwera.

**Formal analysis:** Julia Corey, Hilary Schwandt, Angel Boulware, Ana Herrera, Ethan Hudler, Claudette Imbabazi, Ilia King, Jessica Linus, Innocent Manzi, Madelyn Merrit, Lyn Mezier, Abigail Miller, Haley Morris, Dieudonne Musemakweli, Uwase Musekura, Divine Mutuyimana, Chimene Ntakarutimana, Nirali Patel, Adriana Scanteianu, Biganette-Evidente Shemeza, Gi'anna Sterling-Donaldson, Chantal Umutoni, Lyse Uwera, Madeleine Zeiler, Seth Feinberg.

**Funding acquisition:** Hilary Schwandt, Seth Feinberg.

**Investigation:** Julia Corey, Hilary Schwandt, Angel Boulware, Ana Herrera, Ethan Hudler, Claudette Imbabazi, Ilia King, Jessica Linus, Innocent Manzi, Madelyn Merrit, Lyn Mezier, Abigail Miller, Haley Morris, Dieudonne Musemakweli, Uwase Musekura, Divine Mutuyimana, Chimene Ntakarutimana, Nirali Patel, Adriana Scanteianu, Biganette-Evidente Shemeza, Gi'anna Sterling-Donaldson, Chantal Umutoni, Lyse Uwera, Madeleine Zeiler, Seth Feinberg.

**Methodology:** Hilary Schwandt.

**Project administration:** Hilary Schwandt, Seth Feinberg.

**Resources:** Hilary Schwandt, Seth Feinberg.

**Supervision:** Hilary Schwandt, Seth Feinberg.

**Validation:** Julia Corey, Hilary Schwandt, Angel Boulware, Ana Herrera, Ethan Hudler, Claudette Imbabazi, Ilia King, Jessica Linus, Innocent Manzi, Madelyn Merrit, Lyn Mezier, Abigail Miller, Haley Morris, Uwase Musekura, Divine Mutuyimana, Chimene Ntakarutimana, Nirali Patel, Adriana Scanteianu, Biganette-Evidente Shemeza, Gi'anna Sterling-Donaldson, Chantal Umutoni, Lyse Uwera, Madeleine Zeiler, Seth Feinberg.

**Visualization:** Julia Corey, Hilary Schwandt, Dieudonne Musemakweli.

**Writing – original draft:** Julia Corey, Hilary Schwandt.

**Writing – review & editing:** Julia Corey, Hilary Schwandt, Angel Boulware, Ana Herrera, Ethan Hudler, Claudette Imbabazi, Ilia King, Jessica Linus, Innocent Manzi, Madelyn Merrit, Lyn Mezier, Abigail Miller, Haley Morris, Dieudonne Musemakweli, Uwase Musekura, Divine Mutuyimana, Chimene Ntakarutimana, Nirali Patel, Adriana Scanteianu, Biganette-

Evidente Shemeza, Gi'anna Sterling-Donaldson, Chantal Umutoni, Lyse Uwera, Madeleine Zeiler, Seth Feinberg.

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
