## [Decision Letter · Decision Letter 0]

9 Nov 2021

PONE-D-21-23342Family planning demand generation in Rwanda: Government efforts at the national and community level impact interpersonal communication and family normsPLOS ONE

Dear Dr. Corey,

Thank you for submitting your manuscript to PLOS ONE. After careful consideration, we feel that it has merit but does not fully meet PLOS ONE’s publication criteria as it currently stands. Therefore, we invite you to submit a revised version of the manuscript that addresses the points raised during the review process. PLOS ONE considers qualitative and mixed-methods studies for publication. We recommend that authors use the COREQ checklist, or other relevant checklists listed by the Equator Network, such as the SRQR, to ensure complete reporting (http://journals.plos.org/plosone/s/submission-guidelines#loc-qualitative-research). In general, we would expect qualitative studies to include the following: 1) defined objectives or research questions; 2) description of the sampling strategy, including rationale for the recruitment method, participant inclusion/exclusion criteria and the number of participants recruited; 3) detailed reporting of the data collection procedures; 4) data analysis procedures described in sufficient detail to enable replication; 5) a discussion of potential sources of bias; and 6) a discussion of limitations. Based on the detailed discussions by the reviewers it emerges that methods are insufficiently explained so that replicability is not guaranteed, as well as the rationale and limitations of the study. 

We look forward to receiving your revised manuscript.

Kind regards,

José Antonio Ortega, Ph.D.

Academic Editor

PLOS ONE

Journal Requirements:

2. Please include a copy of the interview guide used in the study, in both the original language and English, as Supporting Information, or include a citation if it has been published previously.

Reviewers' comments:

Reviewer's Responses to Questions

**Comments to the Author**

1. Is the manuscript technically sound, and do the data support the conclusions?

Reviewer #1: Yes

Reviewer #2: Partly

2. Has the statistical analysis been performed appropriately and rigorously? 

Reviewer #1: N/A

Reviewer #2: N/A

3. Have the authors made all data underlying the findings in their manuscript fully available?

Reviewer #1: No

Reviewer #2: No

4. Is the manuscript presented in an intelligible fashion and written in standard English?

Reviewer #1: Yes

Reviewer #2: Yes

5. Review Comments to the Author

Reviewer #1: This is an important topic. The authors conducted a qualitative study to examine the Rwandan government’s efforts to increase the demand of family planning methods and the resulting effect on community-level and individual level perceptions on contraceptives. The paper does make some good contributions but for the lessons from Rwanda to be understood clearly and applied elsewhere, there are some concerns that should be fixed prior to possible publication. It will also be nice if authors can clearly articulate what novel and/or surprising findings they are contributing.

Line 66 please specify year

Line 77 in their conclusion, the authors state that they used the diffusions of innovations theory as a framework of discussing the present work. However, as a reviewer, it was not clear to me from the introduction what this theory is and I was therefore not able to assess properly how it ties into the important topic discussed here.

Line 85 Authors should qualify what kind of messages they are referring to

Line 92 please clarify: past few months relative to what?

Line 95 I understand that the paper is focused on family planning but I think it would be good for the readers to be clear that Umuganda is not a day for family planning as the sentence currently reads. It is a community day that where a range of information is shared including family planning.

Line 97 redundant information on the 5 years. Also authors should clarify whether the contraceptive prevalence is referring to modern contraceptives. Please be sure to specify throughout paper including in abstract. Also after all of the statistics provided, the reader is left wondering what exactly these challenges that have led to the stagnating contraceptive prevalence are. A sentence or 2 providing additional context would suffice. Also something to consider, Rwanda had a new DHS survey (fieldwork during November 2019 - July 2020). The statistics provided may have changed and the fieldwork timeframe of new Rwanda DHS is closer to when authors did their interviews.

Line 104 increase prevalence instead of advance? Also this paragraph needs to be anchored a bit more into the Rwanda context.

Line 121 unclear who the interviewers were, how they were recruited and their training

Line 127 is thematic saturation a method or an endpoint? The authors should revise this sentence for clarity.

Line 175 overwhelmingly positively?

Line 246 revise sentence as the quote is unclear as currently written.

Line 258 the authors should be consistent in the tense they use in their writing. Please revise. Same as line 287.

Line 398 thinking about how these findings may be used elsewhere, how does Rwanda rank in terms of radio use relative to other countries in the region?

Lines 423-429 information is redundant with previous paragraph. It is not clear what this paragraph is specifically adding.

Lines 430 – 436 the authors should engage with the literature and comment whether this has been observed elsewhere

Line 472 why cite only CHWs and leave out nurses here?

Lines 474- 476 Authors should comment somewhere whether findings are as applicable in rural vs urban settings. ie are Musanze and Nyamasheke more rural, urban? It was interesting in the introduction that the authors provided information that contraceptive uptake also increased by 36% among rural women but it would be good to situate the study sites as well.

Reviewer #2: I enjoyed reading this article which addresses an important topic. however, i have a number of concerns about the paper.

1. It is not evident what your research adds to scholarship. this needs to be much clearer. Your methodology does not enable you to demonstrate the impact of the campaign to encourage the uptake of contraception in Rwanda. It only enables you to say what the participants in your research told you about the uptake of contraception. It is also difficult to say anything about the period between 2005-2010. People's memories are faculty and many of your respondents would have been children at the time.

2. You obtained ethical approval from a US university but not in Rwanda. The approval you obtained from the Ministry of Education was for carrying out the research in Rwanda. For health research it is also necessary to have ethical approval from the National Ethics Committee. Were you given an exemption?

3. You have large number of authors whose main contribution was data collection and analysis. However, the interviews were carried out in Kinyarwanda. Do they all speak Kinyarwanda fluently?

4. The methods section is inadequate. It needs to set out how you did your research so that another researcher could replicate it. Things I would have expected to have been includes: development of interview schedules, translation into Kinyarwanda and quality assurance of translations, transcription and translation and quality assurance of transcription, sampling of participants and justification for sampling, data handling and management to ensure confidentiality and anonymity, how the data was analysed and a discussion of any important ethical and safeguarding issues.

5. The paper demonstrates a lack of an understanding of the Rwandan context and provides little contextual information for the reader. There should be a brief introduction to Rwanda so that the reader can understand why the dramatic increase in the uptake of contraction between 2005-2010 is important and of interest.

6. No justification is given for the selection of the two districts beyond that one has a relatively high uptake of contraception and one a low uptake. Why was this important for your research. I would add that the difference is not surprising as Musanze is one of the least poor districts in Rwanda and Nyamasheke by far the poorest district in Rwanda. This for example would account for the differences in hearing information on the radio.

7. You see radio and TV as important in the education of women (Rwanda emphases couples) education. However, in 2005 only 49% of households where at least one female member was under 49 years owned a radio and 3.7% a TV. This had increased to 63% and 8.8% by 2010 and in 2019/20 was 40.4% and 13.6% (RDHS).

8. The first paragraph is confusing because it is not clear what is about 2000 and what comes later. CHWs only become a nationwide service for example in the mid-2000s and you give information on media audiences for 2017 - you could have got data for 2000 etc from the Rwanda Demographic and Health Surveys which can be downloaded from the web site of the National Institute of Statistics website. You could have used the wealth of statistical information in these reports to chart more carefully the changes in uptake of modern contraception among married couples and the decline in the unmet need. Also you could have got statistics for the two districts you sampled.

9. You do not make it clear when you make reference to uptake of contraception if this is all contraception or modern contraception or if it is for all women of child bearing age or only married women. The campaigns have been advocation modern contraception and targeted at married couples.

10. The introduction promises research on why the campaign between 2005 and 2010 was relatively successful but has stalled since. However the empirical research is about the use of contraception in in the late 2010s.

11 You mention the meetings after Umuganda and Akagoro K' ababyeyi as import for dissemination about contraception. Umuganda (community work) is held monthly and has been in place since at least 2000. It is well attended and at the meetings after Umuganda and other village meetings CHWs sensitise the community about contraception among othe health behaviours. Akagoro K' ababyeyi has only been in place since about 2014. The meetings are held once a month for parents and are led by two community volunteers. They are parenting classes and part of the governments strategy for reducing stunting and improving child care and informal education. Contraception is one issue that is discussed.

12 I think that it is premature to say that contraception use is now the norm. There are no recent figures that I am aware of but in 2009 50% of births were unplanned and as you point out and is confirmed by the 2019/2020 RDHS an the increase in uptake of modern contraception stalled after 2010. This may be because ideal family size for both women and men remains an average of just under four children. The Rwanda Government wants to encourage an average of two children. The lack of increase in uptake and use is interesting because since the early 2010s CHWs have been able to dispense modern contraception once a women has been prescribed a contraceptive by a health centre.

13 One of the things that the Rwanda Government consider important in the increase in uptake in modern contraception was an agreement they came to with the churches that they would not encourage married couples not to use modern contraception and the Catholic Church agreed that the Government could have health posts to dispense modern contraception next to Catholic Health Centres. About a third of health centres in Rwanda are run by the Catholic Church.

6. PLOS authors have the option to publish the peer review history of their article (what does this mean?). If published, this will include your full peer review and any attached files.

Reviewer #1: No

Reviewer #2: No

---

## [Author Response · Author response to Decision Letter 0]

28 Feb 2022

Reviewer #1: This is an important topic. The authors conducted a qualitative study to examine the Rwandan government’s efforts to increase the demand of family planning methods and the resulting effect on community-level and individual level perceptions on contraceptives. The paper does make some good contributions but for the lessons from Rwanda to be understood clearly and applied elsewhere, there are some concerns that should be fixed prior to possible publication. It will also be nice if authors can clearly articulate what novel and/or surprising findings they are contributing.

Line 66 please specify year

Thank you for this comment. We have updated the statistics with most recent data and provided the year.

Line 77 in their conclusion, the authors state that they used the diffusions of innovations theory as a framework of discussing the present work. However, as a reviewer, it was not clear to me from the introduction what this theory is and I was therefore not able to assess properly how it ties into the important topic discussed here.

Thank you for highlighting this, and we acknowledge that further expansion on the diffusions of innovation theory was needed. We have added further clarification to the introduction, and removed the mention of its use as a framework from the conclusion.

Line 85 Authors should qualify what kind of messages they are referring to

Thank you for this comment. We have edited this sentence to clarify. 

Line 92 please clarify: past few months relative to what?

Thank you for this comment. We have clarified this sentence. 

Line 95 I understand that the paper is focused on family planning but I think it would be good for the readers to be clear that Umuganda is not a day for family planning as the sentence currently reads. It is a community day that where a range of information is shared including family planning.

Thank you for highlighting this. We have clarified this sentence to be more clear.

Line 97 redundant information on the 5 years. Also authors should clarify whether the contraceptive prevalence is referring to modern contraceptives. Please be sure to specify throughout paper including in abstract. Also after all of the statistics provided, the reader is left wondering what exactly these challenges that have led to the stagnating contraceptive prevalence are. A sentence or 2 providing additional context would suffice. Also something to consider, Rwanda had a new DHS survey (fieldwork during November 2019 - July 2020). The statistics provided may have changed and the fieldwork timeframe of new Rwanda DHS is closer to when authors did their interviews.

Thank you for highlighting this. We have clarified references to contraceptive prevalence throughout the paper. We have also removed the discussion of stagnation, as we do not have data on the topic. Instead, we focus on the continued growth of contraceptive prevalence in Rwanda, based on updated data. 

Line 104 increase prevalence instead of advance? Also this paragraph needs to be anchored a bit more into the Rwanda context.

Thank you for this comment. This paragraph has been revised as suggested. 

Line 121 unclear who the interviewers were, how they were recruited and their training

Thank you for this observation. We have provided additional information to the methods section to clarify this. 

Line 127 is thematic saturation a method or an endpoint? The authors should revise this sentence for clarity.

Thank you for this comment. We have revised this sentence for clarification as suggested. 

Line 175 overwhelmingly positively?

Thank you for noting this oversight. We have amended this language. 

Line 246 revise sentence as the quote is unclear as currently written.

Thank you for noting this – we have removed the part of the quote that was unclear.

Line 258 the authors should be consistent in the tense they use in their writing. Please revise. Same as line 287.

Thank you for noting this oversight. We have amended the tense in both lines referenced to be consistent.

Line 398 thinking about how these findings may be used elsewhere, how does Rwanda rank in terms of radio use relative to other countries in the region?

Thank you for this suggestion, we have edited this paragraph to include discussion of radio use in neighboring countries Tanzania and Uganda. 

Lines 423-429 information is redundant with previous paragraph. It is not clear what this paragraph is specifically adding.

Thank you for highlighting this, we have we have edited line 407 to ensure the different points are clear. 

Lines 430 – 436 the authors should engage with the literature and comment whether this has been observed elsewhere

Thank you for this comment, we have expanded this paragraph to include discussion of similar findings in neighboring countries. 

Line 472 why cite only CHWs and leave out nurses here?

Thank you for this point. We have added nurses here and referenced additional research that discusses further in depth how increased investment in human resources may be valuable to Rwanda’s family planning program.

Lines 474- 476 Authors should comment somewhere whether findings are as applicable in rural vs urban settings. ie are Musanze and Nyamasheke more rural, urban? It was interesting in the introduction that the authors provided information that contraceptive uptake also increased by 36% among rural women but it would be good to situate the study sites as well.

Thank you for this comment. We have added details about differences between the two study districts to the beginning of the methods section.

Reviewer #2: I enjoyed reading this article which addresses an important topic. however, i have a number of concerns about the paper.

1. It is not evident what your research adds to scholarship. this needs to be much clearer. Your methodology does not enable you to demonstrate the impact of the campaign to encourage the uptake of contraception in Rwanda. It only enables you to say what the participants in your research told you about the uptake of contraception. It is also difficult to say anything about the period between 2005-2010. People's memories are faculty and many of your respondents would have been children at the time.

Thank you for this comment. We have expanded on several areas within the discussion and conclusion to make our contributions to the literature clearer. We agree that we cannot conclude that the government’s campaign directly led to the uptake of contraceptives in Rwanda. Instead, we focus on the way that the campaign has impacted discourse and communication among the population, and we suggest that this may offer insights as to why family planning use increased rapidly. We have removed discussion about the period from 2005 to 2010 in relation to findings from this study. 

2. You obtained ethical approval from a US university but not in Rwanda. The approval you obtained from the Ministry of Education was for carrying out the research in Rwanda. For health research it is also necessary to have ethical approval from the National Ethics Committee. Were you given an exemption?

At the time of applying for research approval we were directed to start with the Ministry of Education. The Ministry of Education would then tell us if we needed to proceed to the National Ethics Committee to conduct health research. Given the focus and methods of this research, it was deemed by the Ministry of Education that we need not proceed for health approval. We can provide you with the documentation if you are interested. 

3. You have large number of authors whose main contribution was data collection and analysis. However, the interviews were carried out in Kinyarwanda. Do they all speak Kinyarwanda fluently?

Seven of the co-authors (CI, IM, DM, DM, BS, CU, and LU) are native Kinyarwanda speakers and carried out the FGDs and interviews. Audio recordings and field notes were then transcribed by these authors working alongside native English speakers (all other authors) on site the same day. All authors participated in data analysis, carried out in English. We have added clarification to this point in the methods section, “All audio recordings and field notes were transcribed into English by the interviewers, working alongside native English speakers, on site immediately following data collection.”

4. The methods section is inadequate. It needs to set out how you did your research so that another researcher could replicate it. Things I would have expected to have been includes: development of interview schedules, translation into Kinyarwanda and quality assurance of translations, transcription and translation and quality assurance of transcription, sampling of participants and justification for sampling, data handling and management to ensure confidentiality and anonymity, how the data was analysed and a discussion of any important ethical and safeguarding issues.

Thank you for this comment. We have added details to respond to these requests in the methods section. 

5. The paper demonstrates a lack of an understanding of the Rwandan context and provides little contextual information for the reader. There should be a brief introduction to Rwanda so that the reader can understand why the dramatic increase in the uptake of contraction between 2005-2010 is important and of interest.

Thank you for this suggestion. We have provided additional context to the introduction to this point. 

6. No justification is given for the selection of the two districts beyond that one has a relatively high uptake of contraception and one a low uptake. Why was this important for your research. I would add that the difference is not surprising as Musanze is one of the least poor districts in Rwanda and Nyamasheke by far the poorest district in Rwanda. This for example would account for the differences in hearing information on the radio.

Thank you for this comment, we have added details to the methods section on why these two districts were selected, and highlighted this point in the discussion.

7. You see radio and TV as important in the education of women (Rwanda emphases couples) education. However, in 2005 only 49% of households where at least one female member was under 49 years owned a radio and 3.7% a TV. This had increased to 63% and 8.8% by 2010 and in 2019/20 was 40.4% and 13.6% (RDHS).

Thank you for pointing out the disparity in radio and TV ownership in Rwanda. We have also focused our findings on the impact of the radio due to this fact – and also what has come out in our findings.

8. The first paragraph is confusing because it is not clear what is about 2000 and what comes later. CHWs only become a nationwide service for example in the mid-2000s and you give information on media audiences for 2017 - you could have got data for 2000 etc from the Rwanda Demographic and Health Surveys which can be downloaded from the web site of the National Institute of Statistics website. You could have used the wealth of statistical information in these reports to chart more carefully the changes in uptake of modern contraception among married couples and the decline in the unmet need. Also you could have got statistics for the two districts you sampled.

We have carried out an extensive re-write of the introduction and addressed several of these concerns, providing additional statistics using all RHDS available. 

9. You do not make it clear when you make reference to uptake of contraception if this is all contraception or modern contraception or if it is for all women of child bearing age or only married women. The campaigns have been advocation modern contraception and targeted at married couples.

Thank you for highlighting this. We have clarified references to contraceptive prevalence throughout the paper.

10. The introduction promises research on why the campaign between 2005 and 2010 was relatively successful but has stalled since. However the empirical research is about the use of contraception in in the late 2010s.

Thank you for this important point. We have removed the discussion of stagnation and now focus on the continued growth of contraceptive prevalence in Rwanda, based on updated data from 2020. 

11 You mention the meetings after Umuganda and Akagoro K' ababyeyi as import for dissemination about contraception. Umuganda (community work) is held monthly and has been in place since at least 2000. It is well attended and at the meetings after Umuganda and other village meetings CHWs sensitise the community about contraception among othe health behaviours. Akagoro K' ababyeyi has only been in place since about 2014. The meetings are held once a month for parents and are led by two community volunteers. They are parenting classes and part of the governments strategy for reducing stunting and improving child care and informal education. Contraception is one issue that is discussed.

Thank you for this comment. We have edited our definitions of Umuganda and Akagoroba K’ababyeyi in our results section to make this clearer. 

12 I think that it is premature to say that contraception use is now the norm. There are no recent figures that I am aware of but in 2009 50% of births were unplanned and as you point out and is confirmed by the 2019/2020 RDHS an the increase in uptake of modern contraception stalled after 2010. This may be because ideal family size for both women and men remains an average of just under four children. The Rwanda Government wants to encourage an average of two children. The lack of increase in uptake and use is interesting because since the early 2010s CHWs have been able to dispense modern contraception once a women has been prescribed a contraceptive by a health centre.

Thank you for this point. While contraceptive use among married couples may no longer be taboo, perhaps it is too early to call it the norm; data from 2019 suggest 30% of births among married women were unplanned. As such, we have amended wording in both the results and discussion sections to clarify that contraceptive use is normalized, rather than the norm. 

13 One of the things that the Rwanda Government consider important in the increase in uptake in modern contraception was an agreement they came to with the churches that they would not encourage married couples not to use modern contraception and the Catholic Church agreed that the Government could have health posts to dispense modern contraception next to Catholic Health Centres. About a third of health centres in Rwanda are run by the Catholic Church.

This is a fascinating history. We do not go into these details as the distinction of health posts/health centers and the relation to religious affiliation was not a finding that emerged from our work.

---

## [Decision Letter · Decision Letter 1]

23 Mar 2022

Family planning demand generation in Rwanda: Government efforts at the national and community level impact interpersonal communication and family norms

PONE-D-21-23342R1

Dear Dr. Corey,

We’re pleased to inform you that your manuscript has been judged scientifically suitable for publication and will be formally accepted for publication once it meets all outstanding technical requirements.

Kind regards,

José Antonio Ortega, Ph.D.

Academic Editor

PLOS ONE

Additional Editor Comments (optional):

Both referees and the editor feel that the changes have addressed the issues raised. Congratulations!

Reviewers' comments:

Reviewer's Responses to Questions

**Comments to the Author**

1. If the authors have adequately addressed your comments raised in a previous round of review and you feel that this manuscript is now acceptable for publication, you may indicate that here to bypass the “Comments to the Author” section, enter your conflict of interest statement in the “Confidential to Editor” section, and submit your "Accept" recommendation.

Reviewer #1: All comments have been addressed

Reviewer #2: All comments have been addressed

2. Is the manuscript technically sound, and do the data support the conclusions?

Reviewer #1: Yes

Reviewer #2: (No Response)

3. Has the statistical analysis been performed appropriately and rigorously? 

Reviewer #1: N/A

Reviewer #2: N/A

4. Have the authors made all data underlying the findings in their manuscript fully available?

Reviewer #1: No

Reviewer #2: No

5. Is the manuscript presented in an intelligible fashion and written in standard English?

Reviewer #1: Yes

Reviewer #2: Yes

6. Review Comments to the Author

Reviewer #1: The manuscript is very much improved. Congratulations on this important piece of work. Minor suggestion to add the units of the TFR on line 121. Best wishes!

Reviewer #2: (No Response)

7. PLOS authors have the option to publish the peer review history of their article (what does this mean?). If published, this will include your full peer review and any attached files.

Reviewer #1: No

Reviewer #2: No

---

## [Editor Report · Acceptance letter]

29 Mar 2022

PONE-D-21-23342R1 

Family planning demand generation in Rwanda: Government efforts at the national and community level impact interpersonal communication and family norms 

Dear Dr. Corey:

I'm pleased to inform you that your manuscript has been deemed suitable for publication in PLOS ONE. Congratulations! Your manuscript is now with our production department. 

Kind regards, 

on behalf of

Dr. José Antonio Ortega 

Academic Editor

PLOS ONE